# Clinical and Molecular Implications of Osteopontin in Heart Failure

Argen Mamazhakypov [1] [ID], Meerim Sartmyrzaeva [2], Akpay Sh. Sarybaev [2], Ralph Schermuly [1] and Akylbek Sydykov [1,*] [ID]

1 Department of Internal Medicine, German Center for Lung Research (DZL), Justus Liebig University of Giessen, 35392 Giessen, Germany
2 Department of Mountain and Sleep Medicine and Pulmonary Hypertension, National Center of Cardiology and Internal Medicine, Bishkek 720040, Kyrgyzstan
* Correspondence: akylbek.sydykov@innere.med.uni-giessen.de

**Abstract:** The matricellular protein osteopontin modulates cell–matrix interactions during tissue injury and healing. A complex multidomain structure of osteopontin enables it not only to bind diverse cell receptors but also to interact with various partners, including other extracellular matrix proteins, cytokines, and growth factors. Numerous studies have implicated osteopontin in the development and progression of myocardial remodeling in diverse cardiac diseases. Osteopontin influences myocardial remodeling by regulating extracellular matrix production, the activity of matrix metalloproteinases and various growth factors, inflammatory cell recruitment, myofibroblast differentiation, cardiomyocyte apoptosis, and myocardial vascularization. The exploitation of osteopontin loss- and gain-of-function approaches in rodent models provided an opportunity for assessment of the cell- and disease-specific contribution of osteopontin to myocardial remodeling. In this review, we summarize the recent knowledge on osteopontin regulation and its impact on various cardiac diseases, as well as delineate complex disease- and cell-specific roles of osteopontin in cardiac pathologies. We also discuss the current progress of therapeutics targeting osteopontin that may facilitate the development of a novel strategy for heart failure treatment.

**Keywords:** osteopontin; right ventricle; left ventricle; heart failure





## 1. Introduction

Despite considerable advances in prevention, diagnosis, and treatment, cardiovascular diseases (CVDs) remain the leading cause of death worldwide [1]. Many CVDs ultimately culminate with the development of heart failure (HF). About 64 million patients suffer from HF worldwide, which causes a substantial morbidity and mortality burden on society [2,3]. Approximately half of the HF patients die within five years after diagnosis, and HF remains the leading cause of hospitalization in patients aged over 65 years [3]. The available pharmacotherapies and interventional strategies can only delay HF progression, and a curative treatment approach has not been developed yet.

HF may involve either the left ventricle (LV) or right ventricle (RV), or both. The major CVDs leading to chronic HF affect mostly the left ventricle and include coronary artery disease, arterial hypertension, valvular heart disease, and cardiomyopathies. The main cause of RV failure is pulmonary hypertension (PH), a severe condition characterized by pulmonary vascular remodeling, leading to an increase in pulmonary vascular resistance (PVR) and pulmonary artery pressure (PAP). Consequently, elevated PAP exerts an increased hemodynamic load on the RV, resulting in its structural and functional changes, eventually leading to right HF and premature death. PH is caused by various conditions, including lung diseases, thromboembolic diseases, vascular diseases, and also left HF [4]. In all these conditions, the presence and severity of the RV remodeling and dysfunction determine the adverse outcome. Despite presumably similar processes observed in both

ventricles, such as myocardial fibrosis, cardiomyocyte hypertrophy, capillary loss, myocardial inflammation, etc., pharmacological agents approved for the treatment of LV failure fail to exert benefits in right HF. This can be partly explained by the fact that the signaling pathways activated in left and right HF might significantly differ [5]. Thus, identifying potential mechanisms accounting for these differences and similarities could uncover novel signaling pathways that could be targeted by pharmacological agents.

Myocardial ischemia and chronic pressure or volume overload imposed on the ventricular walls are considered important factors leading to the changes in the heart structure and function. Initially, cardiac injury is characterized by the activation of the neuroendocrine systems, such as the sympathetic nervous system and the renin–angiotensin–aldosterone system, which help to maintain cardiac function [6]. However, continuous injury imposed on the myocardium gradually limits compensatory mechanisms resulting in adverse changes in the heart, decreased relaxation of the myocardium (impaired filling), and deterioration of the pump function (systolic dysfunction) [6]. The pathophysiology of these remodeling processes involves a complex interplay and a network of various intra- and extracardiac cells via direct contact or through various secreted factors [6].

Cardiac remodeling is initially an adaptive rearrangement of the myocardial micro-and macrostructure, which eventually results in adverse alterations in heart performance [6,7]. The mechanisms of cardiac remodeling are complex and remain incompletely understood. However, recent studies have revealed diverse pathological processes that may contribute to the development and progression of cardiac remodeling including augmented myocardial fibrosis [8,9], inflammation [10,11], impaired myocardial capillarization [12,13], dysregulated neurohormonal homeostasis [14,15], altered metabolism [16,17], mitochondrial dysfunction [18,19] and increased production of reactive oxygen species (ROS) [20,21].

One of the adaptive mechanisms of cardiac remodeling in response to pressure overload is ventricular wall thickening, which develops initially to withstand an increased afterload. Cardiac hypertrophy development is based on the augmented synthesis of contractile proteins enabling cardiomyocytes to increase their length and diameter with the main purpose of enhancing cardiomyocyte contractility. Heart adaptation to the pressure overload is accompanied by the development of myocardial fibrosis [8,9], which provides mechanical support to cardiomyocytes to couple their contractility to the increased ventricular wall stress [22]. Cardiac adaptation to pressure overload is also associated with an increased myocardial capillary density to ensure a proper supply of nutrients and oxygen to the cardiomyocytes [12]. Cardiomyocyte hypertrophy, myocardial fibrosis, and changes in myocardial capillarization are the main components of cardiac remodeling, which occur concurrently and involve intricate overlapping signaling pathways in various heart cells, including cardiomyocytes, fibroblasts, immune cells, and coronary artery endothelial cells through complex crosstalk mediated by various growth factors, cytokines, and chemokines [8,12,23–26]. When taken together, adaptive cardiac remodeling involves initial changes in the ventricular structure aimed at maintaining cardiac function. At some point in the disease evolution, the ventricles lose their ability to withstand sustained pressure overload due to cardiomyocyte apoptosis, inadequate/excessive myocardial fibrosis, and impaired myocardial capillarization. Thus, initially, adaptive heart hypertrophy eventually evolves into maladaptive cardiac remodeling, triggering the transition to HF [6].

The extracellular matrix (ECM) is a collagen-based network occupying extracellular space that provides assembly, support, and organization of the tissues [25]. Myocardial ECM provides structural organization to the cardiac tissue and facilitates electrical conduction, cellular signaling, and intercellular communication [24,25]. Matricellular proteins are considered a family of non-structural ECM proteins capable of interacting with cell surface receptors, growth factors, proteases, and other inflammatory mediators, as well as with structural matrix proteins within the ECM. Accumulating evidence suggests that various matricellular proteins are involved in complex interactions between cardiac cells and ECM to regulate hypertrophic, fibrotic, and inflammatory processes by exerting various cytokine- and chemokine-like effects on cardiac cells in homeostatic as well as stressed

conditions [26–30]. In the myocardium, like in other tissues, matricellular proteins are regulated in response to various stress conditions such as injury, inflammation, and mechanical stretch [31]. Among these matricellular proteins, osteopontin has recently gained increased attention as a key regulator of myocardial injury and repair.

## 2. Osteopontin Biology

Human osteopontin protein consists of 314 amino acid residues with a predicted molecular weight of 32 kilo Daltons (kDa). However, substantial posttranslational modifications, including glycosylation and phosphorylation, increase its actual molecular weight up to 75 kDa on sodium dodecyl sulfate-polyacrylamide gel electrophoresis [32,33]. Mammals display a high degree of homology in the osteopontin coding sequence. Osteopontin is expressed in various cell types, including activated macrophages and T cells, osteoclasts, hepatocytes, smooth muscle, endothelial, and epithelial cells [34]. In the heart, several cell types, such as cardiomyocytes, cardiac fibroblasts, resident macrophages, and coronary artery endothelial cells, express osteopontin in response to diverse stimuli, including hypoxia, inflammation, toxin exposure, and mechanical stretching [31].

Osteopontin is also known as secreted phosphoprotein 1 and early T-lymphocyte activation protein 1, highlighting its role in the inflammation processes and its interactions with the immune system. In addition, it is involved in diverse physiological and pathological processes such as cell adhesion, migration, proliferation, and tissue repair via integrin or CD44 receptors mediated activation of various cellular signaling pathways and interaction with growth factors, cytokines, chemokines, and proteases [35]. Unlike rodents, in humans, alternative splicing of osteopontin results in three isoforms, osteopontin-a (the full-length isoform), osteopontin-b (which lacks exon 5), and osteopontin-c (which lacks exon 4) [36]. Of note, studies investigating the involvement of osteopontin in cardiac pathologies primarily focus on total osteopontin expression and do not consider expression levels of any specific osteopontin isoform [36].

### 2.1. Osteopontin Cleavage

Effects of osteopontin depend not only on its posttranslational modifications [37] but also on its cleavage. Various extracellular proteases including matrix metalloproteinase-3, -7, -9 (MMP-3, -7, -9), and thrombin cleave osteopontin protein at specific sites (Figure 1) [38–40]. For example, MMP-9 cleaves osteopontin at amino acid positions 151–152, 193–194, and 195–196, yielding four peptides with distinct biological activities [38]. Thrombin cleaves osteopontin at the $^{168}$R/S$^{169}$ site into two fragments; one of them is an N-terminal fragment, which exposes a cryptic integrin-binding motif $^{158}$GRGDSVVYGLR$^{168}$ on the human protein (in mice, SLAYGLR). This fragment exerts specific biological effects by binding to $\alpha4\beta1$ and $\alpha9\beta1$ integrins [39,40]. Combined effects of thrombin, MMPs, plasmin, and cathepsin D yield seven N-terminal fragments of osteopontin containing the $^{158}$GRGDSVVYGLR$^{168}$ motif [41]. Thus, osteopontin cleavage is a prerequisite for exposing some receptor binding domains, consequently influencing the manifestation of its diverse biological activities.

### 2.2. Osteopontin Receptors

The majority of the effects of osteopontin are based on its ability to engage with various integrins and CD44 with its specific receptor binding domains (Figure 1) [35]. The thrombin-cleaved N-terminal fragment of osteopontin binds to several integrins, including $\alpha4\beta1$-, $\alpha4\beta7$-, $\alpha5\beta1$-, $\alpha9\beta1$-, $\alpha v\beta1$-, $\alpha v\beta3$- $\alpha v\beta5$-, $\alpha v\beta6$-, and $\alpha5\beta1$ via its two closely located but distinct motifs RGD and SVVYGLR (in mice, SLAYGLR) within the sequence $^{158}$GRGDSVVYGLR$^{168}$ of receptor binding domains [33,42–48]. Specifically, osteopontin interacts with $\alpha v\beta1$, $\alpha v\beta3$, and $\alpha v\beta5$ integrin receptors via the classical RGD sequence, while it interacts with $\alpha9\beta1$, $\alpha4\beta1$, $\alpha4\beta7$ via SVVYGLR [42,49,50]. With the ELVTDFPTDL-PAT motif (in humans), osteopontin binds $\alpha4\beta1$ integrin [51]. Thrombin cleavage is not required for the full adhesion to $\alpha v\beta3$-, $\alpha v\beta5$-, or $\alpha v\beta6$-integrins because they bind to the RGD domain of full osteopontin [44]. Via the C-terminal fragment calcium-binding site

osteopontin interacts with the CD44 splice variants, CD44v3, CD44v6, and CD44v7 [52,53]. Moreover, osteopontin interacts with various ECM proteins such as fibronectin and collagen types I, II, III, IV, and V [54,55]. In summary, osteopontin binds to a complex system of integrin receptors along with CD44 with its receptor binding domains and the overall effect of osteopontin on cell functions may be mediated by the activation of several receptor signaling pathways.

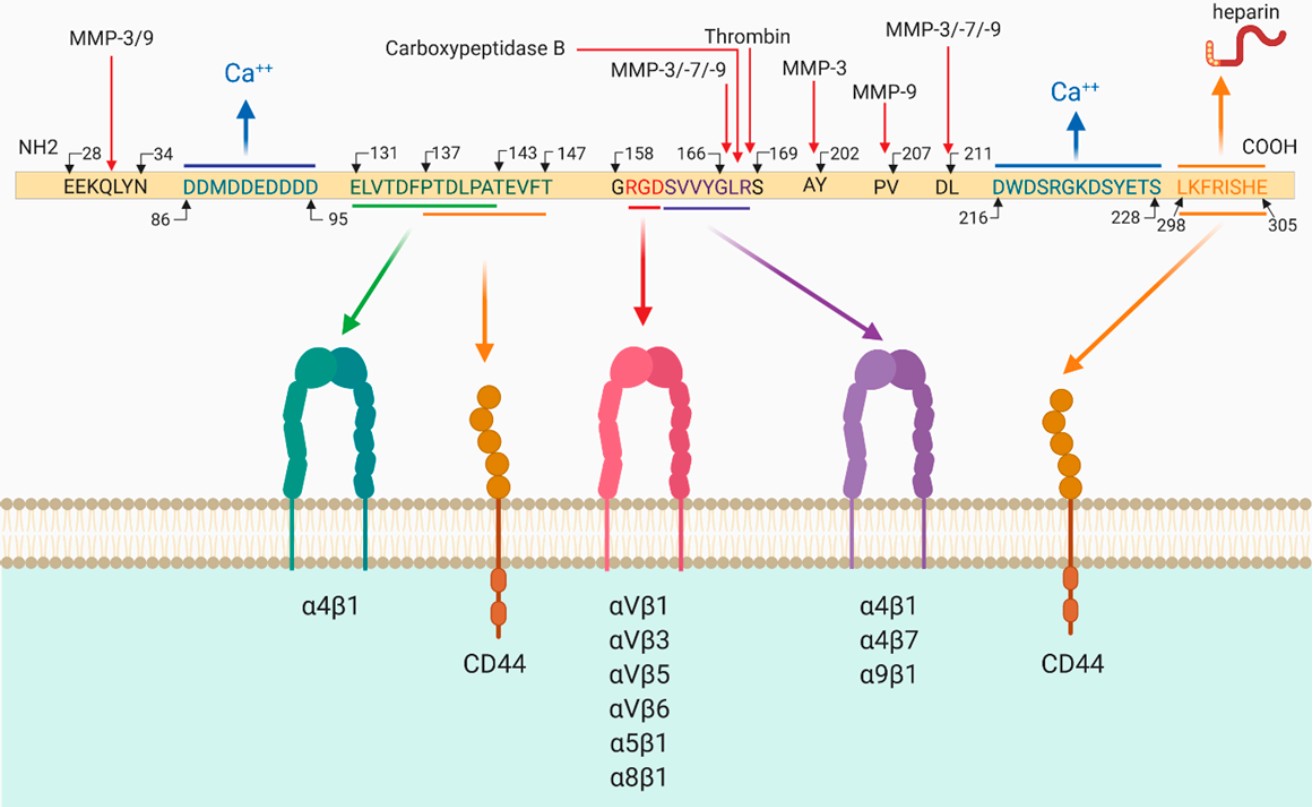

**Figure 1.** Osteopontin biology. Human osteopontin consists of 314 amino acid residues. Osteopontin can be cleaved at several sites by various proteases including thrombin (between amino acids 168-169), matrix metallopeptidase 3 (MMP-3) (between amino acids 31–32, 166–167, 201–202, and 210–2011), MMP-7 (between amino acids 166–167 and 210–211) and MMP-9 (between amino acids 31–32, 166–167, 206–207 and 210–211). ELVVTDFPTDLPAT sequence binds to integrin α4β1. RGD sequence binds to αVβ1, αVβ3, αVβ5, αVβ6, α5β1, α8β1. SVVYGLR sequence binds to α4β1, α4β7 and α9β1. PTDLPATEVFT and LKFRISHE sequences bind to the CD44 receptor.

## 3. Osteopontin in Heart Failure

Embryogenesis is associated with the augmented expression of osteopontin in various tissues and organs, including the heart [56–58]. Despite this implied function during embryogenesis, mice lacking osteopontin grow to maturity without any overt signs of morphological and functional cardiac abnormalities [29,59]. In line with this, in homeostatic conditions, osteopontin expression in the adult heart is very low [58,60–63], although it is regulated in a cell-specific manner. For example, a low basal expression of osteopontin was reported in cardiomyocytes [64], while it can be readily detected in coronary artery endothelial cells and cardiac fibroblasts [64,65]. Nevertheless, osteopontin-null mice respond to a variety of pathological stimuli, including tissue injury, inflammation, and infection, differently from wild-type mice [66].

In the last two decades, several studies utilizing osteopontin depletion either by neutralizing antibodies or by targeted mutagenesis have further advanced our understanding of the role of osteopontin in various cardiac pathologies [59,67–72]. Osteopontin-null mice were used in a number of HF models, including angiotensin-II (Ang-II) infusion [59], al-

dosterone infusion [67], transverse aortic constriction (TAC) [68], desmin-deficient model of dilated cardiomyopathy (DCM) [69], streptozotocin-induced model of diabetic cardiomyopathy [70], left anterior descending artery (LAD) ligation as a model of myocardial infarction (MI) [71] and a brief, repetitive LAD-occlusion model of ischemia-reperfusion (IR)-induced myocardial injury [72]. However, these and other studies have yielded conflicting results suggesting disease- and cell-specific roles of osteopontin in cardiac pathologies.

### 3.1. Cell-Specific Regulation of Osteopontin

#### 3.1.1. Osteopontin in Cardiomyocytes

Several factors have been shown to induce osteopontin expression in cardiomyocytes including aldosterone [73], dexamethasone [64], endothelin-1 [58], norepinephrine (NE) [58], and ROS [74] (Figure 2). In contrast, other known stimulators of osteopontin expression such as Ang-II [65,75], interleukin-1β (IL-1β), and interferon-γ (IFN-γ) [64] failed to induce osteopontin expression in cardiomyocytes.

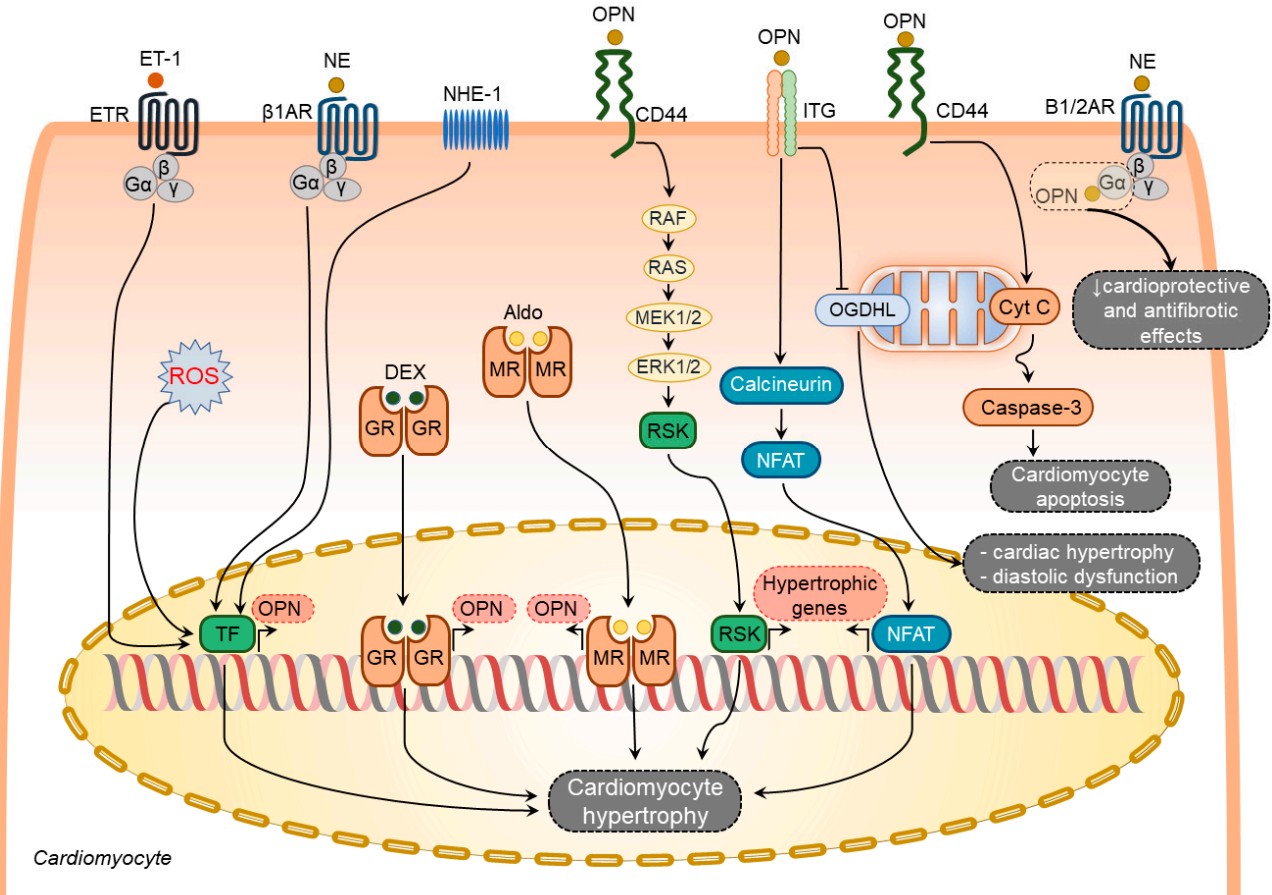

**Figure 2.** Osteopontin signaling in cardiomyocytes. Various factors such as endothelin-1 (ET-1), norepinephrine (NE), sodium-hydrogen exchanger isoform-1 (NHE-1), reactive oxygen species (ROS), dexamethasone (DEX), and aldosterone (Aldo) upregulate osteopontin (OPN) in cardiomyocytes. In turn, osteopontin regulates several processes in cardiomyocytes via modulating certain signaling pathways. These pathways include induction of hypertrophic genes expression via integrin- and CD44-mediated signaling pathways, cardiomyocyte hypertrophy and diastolic dysfunction via blocking mitochondrial protein oxoglutarate dehydrogenase-like (OGDHL), and cardiomyocyte apoptosis via mitochondrial apoptosis signaling pathway, attenuation of the beneficial effects of adrenergic signaling by directly binding and blocking G-α subunit of the adrenergic receptors. The overall effect of osteopontin signaling in cardiomyocytes results in cardiomyocyte hypertrophy, cardiomyocyte apoptosis, and cardiomyocyte mitochondrial dysfunction.

Endothelin-1 and NE-mediated osteopontin upregulation in rat cardiomyocytes coincide with the induction of atrial natriuretic peptide expression [58], suggesting an active involvement of osteopontin in hypertrophic responses of the cardiomyocytes. Cardiomyocyte hypertrophy, LV dilatation, and dysfunction caused by $Na^+/H^+$ exchanger isoform 1 (NHE1) overexpression in mice were associated with osteopontin upregulation and osteopontin-CD44 mediated activation of p90-ribosomal S6 kinase signaling pathways [76]. All these pathological changes were reversed in double transgenic mice expressing active NHE1 and osteopontin knockout [76]. Hypertrophic effects of osteopontin on cardiomyocytes can also be mediated through the calcineurin/NFAT pathway [77,78]. In addition, osteopontin can diminish the cardioprotective and antifibrotic effects of β1- and β2-adrenoceptors signaling pathways by direct binding to their Gαs subunits [73].

Several studies have demonstrated an association of increased osteopontin expression with enhanced cardiomyocyte apoptosis in different HF models [29,67,70]. Cardiomyocyte-specific osteopontin overexpression in mice was associated with decreased LV function and increased cardiomyocyte apoptosis [79]. In support of these findings, treatment of adult rat cardiomyocytes with purified osteopontin or adenoviral-mediated expression of osteopontin induced cell apoptosis via CD44-mediated activation of the mitochondrial death pathway and endoplasmic reticulum stress [79].

Col4a3-deficiency in mice recapitulates multiple features of HF with preserved ejection fraction and is associated with cardiac hypertrophy, diastolic dysfunction, myocardial fibrosis, and mitochondrial dysfunction [80]. Targeting osteopontin significantly ameliorated the HF phenotype in these mice [80]. This study revealed that osteopontin induced mitochondrial dysfunction in cardiomyocytes by downregulating myocardial 2-oxoglutarate dehydrogenase-like, a vital protein in normal mitochondrial function [80].

When taken together, osteopontin serves as an upstream and downstream of diverse signaling pathways involved in cardiomyocyte hypertrophy, apoptosis, and mitochondrial dysfunction (Figure 2).

### 3.1.2. Osteopontin in Cardiac Fibroblasts

In DCM patients, increased levels of myocardial osteopontin strongly correlated with collagen expression [81], suggesting osteopontin involvement in cardiac fibrogenesis. Plasma from aged osteopontin-null mice failed to cause age-specific activation of cardiac fibroblasts [82]. Similarly, cardiac fibroblasts isolated from osteopontin-null mice displayed decreased proliferation and adhesion to ECM [59,83], less spreading, less resistance to detachment by shear stress, and a reduction in collagen gel contraction, which could be partially restored by ectopic osteopontin expression [83].

Several growth factors and cytokines have been shown to regulate osteopontin expression in cardiac fibroblasts, including Ang-II [65,75] and syndecan-4 [84] (Figure 3). Ang-II enhances osteopontin expression in cardiac fibroblasts via NADPH-ROS-mediated activation of ERK1/2 and JNKs pathways [65]. Syndecan-4 increases osteopontin expression via calcineurin/NFAT signaling pathways [84]. Mechanical stretch can also induce osteopontin expression in cardiac fibroblasts [85].

Transforming growth factor-β (TGF-β) is a multifunctional cytokine mediating myofibroblast transformation and collagen synthesis. TGF-β1 failed to induce myofibroblast differentiation of osteopontin-depleted fibroblasts [83], suggesting a pivotal role of osteopontin in myofibroblast differentiation. Osteopontin operates along with high mobility group box 1 intracellularly to form focal adhesions in the myofibroblasts in response to TGF-β1 stimulation [83].

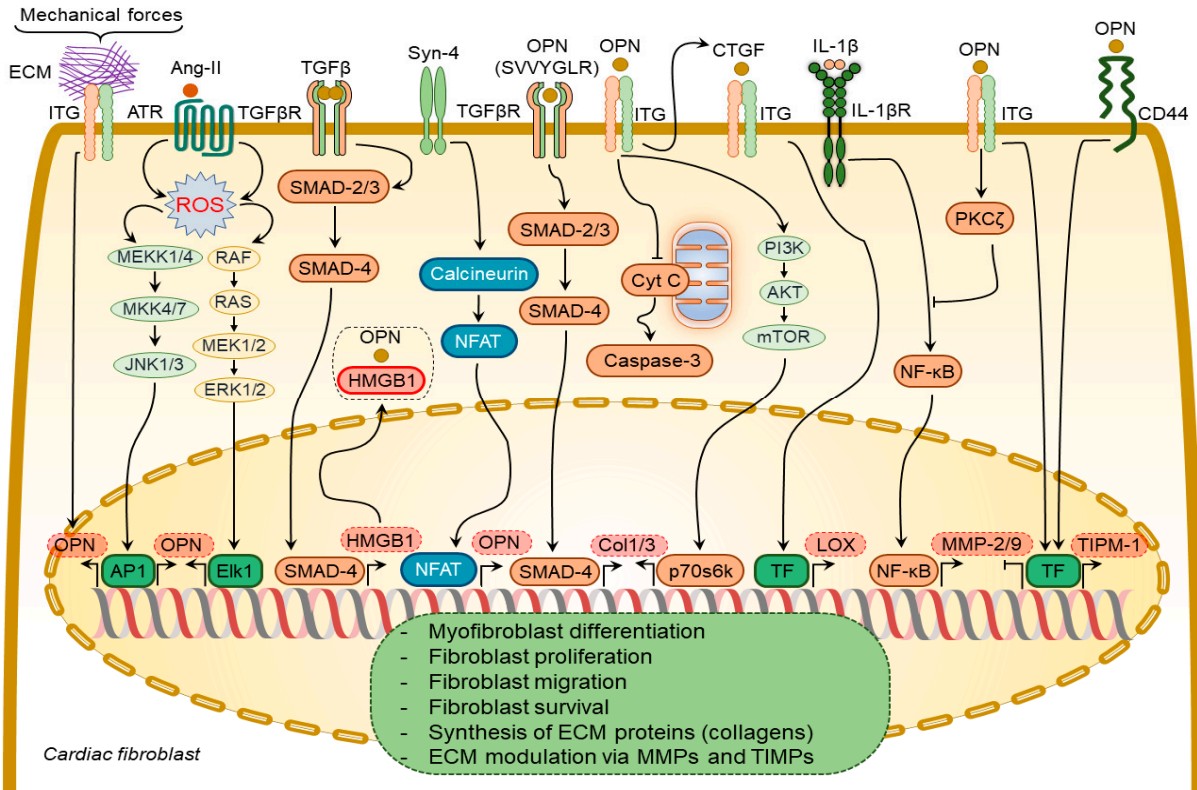

**Figure 3.** Osteopontin signaling in cardiac fibroblasts. A number of factors induce osteopontin (OPN) expression in cardiac fibroblasts including angiotensin-II (Ang-II), syndecan-4 (Syn-4), and mechanical forces. In turn, osteopontin regulates several processes in cardiac fibroblasts via modulating certain signaling pathways. These processes include binding of osteopontin to high mobility group box 1 (HMGB1) to form focal adhesions in cardiac fibroblasts; TGFβR activation via its SVVYGLR motif and subsequent induction of collagen-1 and -3 (Col1/3) expression; enhancement of Col1/3 expression via integrin receptor-mediated signaling pathways; inhibition of cardiac fibroblast apoptosis via blocking the release of cytochrome C (Cyt C) from mitochondria; inhibition of the activity and expression of MMP-2/9 induced by interleukin-1β (IL-1β) signaling pathway; induction of tissue inhibitor of metalloproteinases 1 (TIMP-1) expression via integrin- and CD44-mediated signaling pathways. The overall effect of osteopontin signaling in cardiac fibroblasts results in myofibroblast differentiation, proliferation, migration, and survival, synthesis of extracellular matrix (ECM) proteins, and modulation of ECM via regulating expression of MMPs and TIMPs.

Osteopontin causes collagen-I synthesis and secretion in fibroblasts through a focal adhesion kinase (FAK)/protein kinase B (Akt)-dependent pathway [86]. The thrombin-cleaved N-terminal osteopontin fragment can induce collagen-I and -III expression in cardiac fibroblasts via activation of the TGFβR signaling pathway [85]. Ang-II-induced cardiac fibroblast proliferation and contraction are mediated by osteopontin RGD domain-β3-integrin receptor signaling pathway [87]. A combination of osteopontin and Ang-II promoted the contraction of three-dimensional collagen gels and cardiac fibroblast growth via β3 integrin-mediated signaling pathways [87].

Osteopontin inhibited the expression and activity of MMP-2 and MMP-9 in cardiac fibroblasts induced by IL-1β stimulation [88]. These effects were mediated by β3 integrins-induced activation of the PKC-ζ signaling pathway [88]. Osteopontin upregulated tissue inhibitor matrix metalloproteinase 1 (TIMP-1) and downregulated MMP-1 expression in fibroblasts via αvβ3 and CD44-receptor signaling [89]. Remarkably, MMP-9-cleaved osteopontin fragments containing the RGD motif induced more pronounced cardiac fibroblast migration than the full-length osteopontin [38].

Osteopontin is a strong regulator of lysyl oxidase expression and activity in cardiac fibroblasts, which is responsible for the formation of cross-linked, insoluble collagen and increased ECM accumulation and myocardial stiffness [84,90]. It induces lysyl oxidase upregulation in cardiac fibroblasts [90] via increasing connective tissue growth factor expression [83].

### 3.1.3. Osteopontin in Cardiac Endothelial Cells

Various growth factors and inflammatory mediators have been shown to regulate osteopontin expression in cardiac endothelial cells [64,65,75] (Figure 4). Ang-II increased osteopontin expression in cardiac endothelial cells [65,75] through NADPH-ROS-mediated activation of the Erk1/2 signaling pathway [75]. A combination of IL-1β and IFN-γ also augmented osteopontin expression in cardiac endothelial cells [64]. Moreover, dexamethasone significantly increased osteopontin expression in in vitro experiments [53].

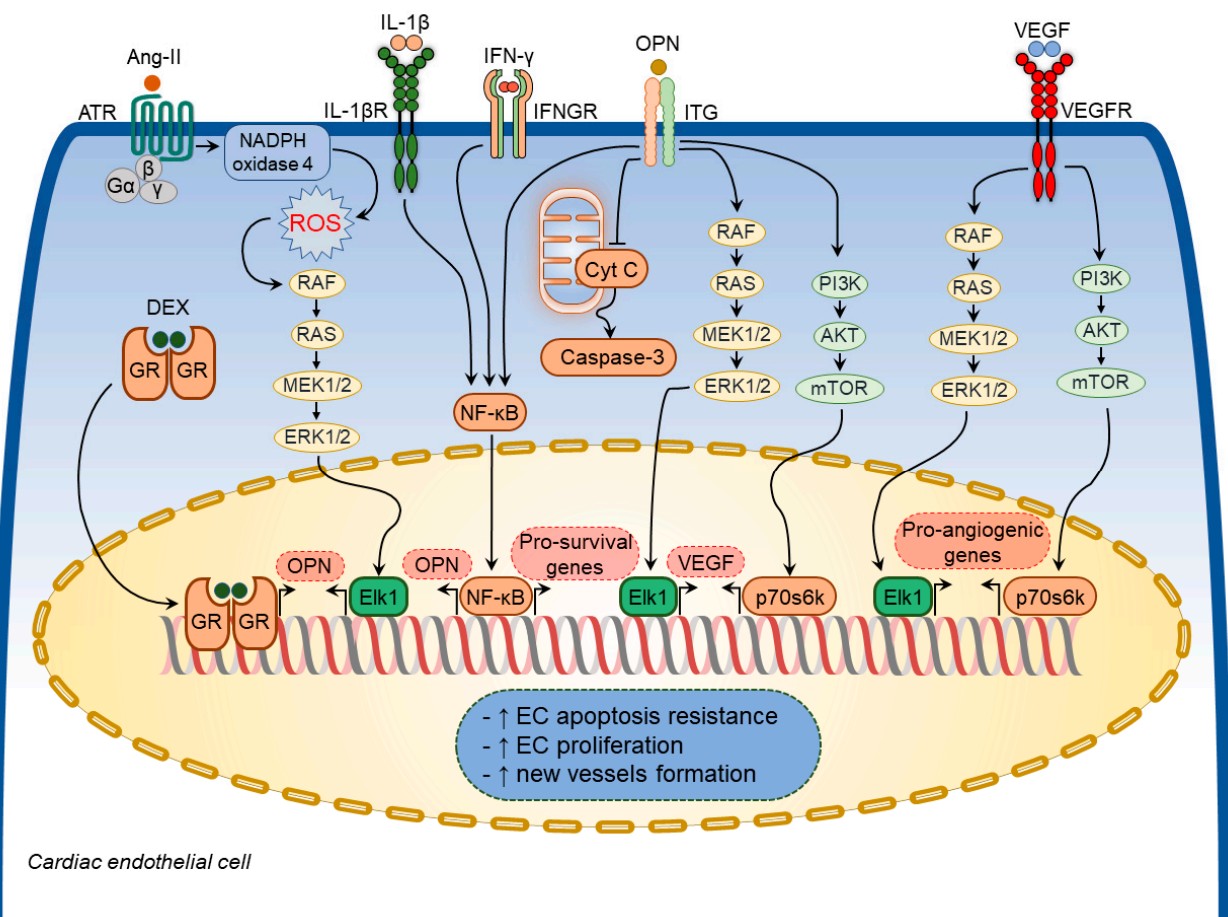

**Figure 4.** Osteopontin signaling in cardiac endothelial cells. A number of factors induce osteopontin (OPN) expression in cardiac endothelial cells including angiotensin-II (Ang-II), interleukin-1β (IL-1β), interferon-γ (IFN-γ), and dexamethasone (DEX). In turn, osteopontin regulates several cellular processes in cardiac endothelial cells via modulating certain signaling pathways. These pathways include osteopontin-induced vascular endothelial growth factor (VEGF) expression and subsequent VEGF-mediated expression of pro-angiogenic factors, inhibition of cardiac endothelial cell apoptosis via blocking mitochondria-mediated apoptosis pathway. Osteopontin signaling in cardiac endothelial cells results in increased apoptosis resistance, increased cell proliferation, and new vessels formation.

Osteopontin exerted cytoprotective effects on endothelial cells and promoted angiogenesis by enhancing vascular endothelial growth factor (VEGF) expression through the phosphatidylinositol 3-kinase (PI3K)/protein kinase B (Akt)- and ERK-mediated path-

ways [91]. In addition, osteopontin facilitated the survival of stressed endothelial cells [92]. Endothelial cells plated on osteopontin-coated surfaces become apoptosis-resistant to serum deprivation via integrin-mediated activation of the nuclear factor kappa B (NF-κB) pathway [93]. Similarly, soluble osteopontin inhibited apoptosis of endothelial cells deprived of essential growth factors [92]. Osteopontin also promoted endothelial regeneration of the injured vessels through activation of RGD-αvβ3 signaling pathways [94]. In line with these findings, osteopontin-null mice showed impaired myocardial angiogenic response post-MI, which resulted in adverse LV remodeling, suggesting that osteopontin plays a crucial role in maintaining and restoring myocardial capillarization following MI [95]. Thus, osteopontin regulates coronary microvascular homeostasis by exerting cytoprotective and pro-survival effects on endothelial cells (Figure 4).

### 3.1.4. Osteopontin in Cardiac Macrophages

Osteopontin plays a key role in macrophage biology by regulating their migration, survival, phagocytosis, and cytokine and chemokine synthesis [96]. It is undetectable in circulating monocytes but is dramatically upregulated during macrophage activation and differentiation, serving as one of the markers of the activated macrophages [97]. Macrophage-specific osteopontin regulation was identified in several experimental HF models [62,63,98]. Thus, osteopontin expression was markedly upregulated in macrophages within and around myocardial injury induced by a transdiaphragmatic freeze-thaw [63] and LAD occlusion models of MI [98]. Similarly, osteopontin was significantly increased in macrophages infiltrating the myocardium in hamsters with heritable cardiomyopathy [62].

Various factors, such as IL-10 and macrophage colony-stimulating factor (M-CSF), regulate osteopontin expression in cardiac macrophages during myocardial injury [99,100]. An essential role of the IL-10-STAT3-galectin-3 axis for osteopontin expression was demonstrated in cardiac macrophages after MI [99]. Interestingly, IL-10 and M-CSF acted synergistically to activate STAT3 and ERK in cardiac macrophages, which in turn upregulated the expression of galectin-3 and MerTK [100]. Activation of these signaling pathways led to the differentiation and functional maturation of osteopontin-producing macrophages resulting in optimal wound healing of the injured myocardium following MI [100]. Any alteration in these pathways can lead to adverse myocardial remodeling following MI [99]. When taken together, osteopontin plays a crucial role in maintaining the reparative properties of cardiac macrophages during myocardial injury.

### 3.2. Implications of Osteopontin in Specific Pathological Processes

### 3.2.1. Osteopontin in Cardiac Hypertrophy

Cardiac hypertrophy is an adaptive response to hemodynamic load characterized by the thickening of the heart walls and enlargement of the individual cardiomyocytes, which has a compensatory role in maintaining cardiac performance and attenuating ventricular wall stress and chamber dilatation. At the cellular level, cardiomyocyte hypertrophy is characterized by an increase in cell dimensions mainly due to augmented contractile protein synthesis [101]. However, sustained hemodynamic stress in pathological conditions such as hypertension and valvular heart disease drives adverse alterations in the myocardium characterized by excessive cardiomyocyte apoptosis, myocardial inflammation, and fibrosis, eventually culminating in ventricular dilation and dysfunction.

Robust upregulation of myocardial osteopontin expression upon mechanical stress was clearly shown in several rodent models [58,68,85]. Pressure overload induced an acute and strong elevation of osteopontin expression in the myocardium in rodents following TAC [58,68,85] and in rats with renovascular hypertension [58]. The increase in myocardial osteopontin expression was reported to be proportional to the degree of the afterload [102]. In line with these findings, there was a strong correlation between expression levels of osteopontin and atrial natriuretic peptide in the LV myocardium in renovascular and TAC-induced LV remodeling models [58]. In the latter models, cardiomyocytes were the main cell type in the heart producing osteopontin [58]. In contrast, in spontaneously hypertensive

rats with cardiac hypertrophy, osteopontin expression was identified primarily in non-myocytes in the interstitial and perivascular space (possibly infiltrating macrophages and fibroblasts) [103].

The exploitation of osteopontin-null mice in several animal models of cardiac hypertrophy revealed that osteopontin is a key mediator in the mechanical stress-induced myocardial hypertrophic response [59,68,76,80]. In mice subjected to TAC, a lack of osteopontin was associated with attenuated cardiac hypertrophic response [68]. Knocking out osteopontin in transgenic mice with NHE1 overexpression significantly reduced cardiac hypertrophy, attenuated collagen deposition, and improved cardiac function [76]. Similarly, in another genetic mouse model of HF with preserved ejection fraction induced by Col4a3 deficiency, osteopontin deletion was associated with improved parameters of LV diastolic function and cardiac hypertrophy reduction [80]. Controversial data were obtained in the model of Ang-II infusion [29,59]. Thus, in one study, osteopontin deficiency attenuated cardiac hypertrophy induced by Ang-II infusion [59], whereas, in another study, osteopontin deletion did not prevent cardiac hypertrophy, despite attenuated myocardial fibrosis [29].

### 3.2.2. Osteopontin in Cardiac Inflammation

Heart diseases are associated with inflammation, which contributes to cardiac dysfunction and myocardial fibrosis [10,104,105]. Numerous cytokines and chemokines are actively involved in the onset and progression of myocardial fibrosis [10,105]. Osteopontin has been shown to exert cytokine- and chemokine-like functions [35].

Among various inflammatory cells, macrophages represent the dominant inflammatory cell population in the remodeled myocardium [106]. Macrophage infiltration of the myocardium has been shown in various models, including spontaneously hypertensive rats and murine TAC [106,107]. Complete abolition of CD45+ monocytes recruitment to the myocardium in response to Ang-II infusion in osteopontin-null mice [108] suggested the involvement of osteopontin in this process.

Osteopontin expressed in circulating leucocytes may also be actively involved in inflammatory cell recruitment to the myocardium and systemic inflammation in patients with various cardiac diseases [109]. Circulating CD4+ T lymphocytes expressing osteopontin and circulating osteopontin levels both correlate with the New York Heart Association Functional Class (NYHA FC) in HF patients and are associated with plaque instability in coronary artery disease patients [109,110].

Inflammatory cells have been identified as one of the major contributors to fibrogenesis in different tissues and organs under various disease conditions [111]. During the acute inflammatory phase of wound healing processes upon tissue injury in different tissues and organs, osteopontin expression is enhanced in infiltrating leukocytes, while during the chronic inflammatory phase, its upregulation is observed in resident macrophages [34]. Increased osteopontin expression by resident and recruited inflammatory cells increases collagen deposition and accumulation in the tissues [109,110,112].

Osteopontin expression by immune/inflammatory cells is associated with cardiac hypertrophic and fibrotic responses in the settings of a number of heart diseases [62,63,69,108,113]. Studies exploiting mice with a genetically modified osteopontin expression provided important insights into its role in myocardial inflammation in diverse cardiac conditions [69,79,112,114–117]. Mice overexpressing osteopontin in cardiomyocytes spontaneously develop severe cardiomyopathy, which is characterized by enhanced recruitment of inflammatory cells to the myocardium and excessive collagen accumulation, subsequently leading to chronic myocarditis and eventually premature death [79,112]. In line with these reports, deletion of osteopontin in desmin deficient mice, which spontaneously develop DCM with increased myocardial inflammation and fibrosis, was associated with an attenuation of myocardial inflammation and improvement in LV systolic function [69].

Cardiomyocyte-specific integrin-linked kinase (ILK) deficient mice spontaneously develop lethal cardiomyopathy characterized by excessive inflammatory cell accumulation, myocardial fibrosis, and cardiomyocyte apoptosis [115]. Augmented osteopontin expression in cardiomyocytes revealed by comprehensive profiling and mitigation of HF severity in these mice by application of anti-osteopontin antibodies implicated osteopontin as a major contributor to this phenotype [115]. Likewise, in other murine models, including those of Chagas heart disease and viral myocarditis, osteopontin-null mice displayed improved cardiac remodeling and attenuated myocardial inflammation [116] [117].

When taken together, osteopontin is highly upregulated in acute and chronic inflammatory phases of the myocardial injury and can contribute to adverse myocardial remodeling and cardiac dysfunction by promoting the recruitment of inflammatory cells to the myocardium.

### 3.2.3. Osteopontin in Cardiac Fibrosis

Osteopontin induces collagen expression in several cell types, and its expression is closely associated with the collagen deposition and tissue fibrosis of multiple organs, including the liver, kidney, lung, and heart [71,90,108,113,118–120], suggesting a pathogenic role of osteopontin in fibrotic processes in a variety of tissues and organs.

Osteopontin is involved in reparative processes after MI by promoting myocardial fibrosis and thus preventing post-MI ventricular chamber dilatation and systolic dysfunction [60,71]. Recent evidence suggests that osteopontin-producing macrophages are important players in mediating these effects by removing tissue debris and stimulating collagen synthesis [63,99,121]. The beneficial effects of osteopontin may also be explained by its ability to inhibit MMP-2 and MMP-9 activation after MI [122]. In contrast, in the MDX mouse model of Duchenne muscular dystrophy and associated DCM, osteopontin was involved in myocardial fibrosis and cardiac dysfunction via MMP-9 upregulation [123].

The most prominent phenotype of osteopontin-null mice in various HF models is attenuated myocardial fibrosis [124,125]. Osteopontin deficiency prevented fibrotic responses in a number of rodent models, including Ang-II infusion [29], aldosterone infusion [67], and LAD occlusion [71]. However, the attenuation of myocardial fibrosis in the osteopontin-null mice was associated with more severe impairment of cardiac systolic function and prominent ventricular dilation compared to wild-type counterparts [29,67,71]. The underlying mechanisms, though, remain elusive. Contrary to these studies, osteopontin-null mice subjected to TAC displayed a degree of myocardial fibrosis, which was comparable to that of wild-type counterparts [68]. In another study, osteopontin-null mice subjected to TAC developed more severe myocardial fibrosis along with deteriorated LV dysfunction [126]. Nevertheless, targeting osteopontin with a specific aptamer was beneficial in treating LV remodeling and dysfunction in wild-type TAC mice [126], suggesting that complete loss of osteopontin might lead to more severe cardiac dysfunction due to either attenuated or exaggerated myocardial fibrosis, whereas partial osteopontin blocking with pharmacological agents (aptamers) seems to be beneficial.

In contrast to the aforementioned studies, showing an association of decreased myocardial fibrosis with heart function deterioration in osteopontin deficiency, in the streptozotocin-induced diabetic cardiomyopathy model, attenuated myocardial fibrosis in osteopontin-null mice was associated with improved LV function [70]. Similarly, in mice with HF due to desmin deficiency, osteopontin deletion ameliorated HF severity at least in part due to mitigation of myocardial fibrosis [69]. Improved cardiac function along with reduced collagen deposition following osteopontin deletion was further demonstrated in a genetic model of cardiac hypertrophy due to NHE1 overexpression [76]. Furthermore, inhibition of osteopontin with shRNA suppressed myocardial fibrosis resulting in improvement of cardiac remodeling and function in a mouse model of DCM due to expression of the mutant troponin [86]. Thus, in several predominantly genetic models of HF, suppressed fibrotic response in mice lacking osteopontin is associated with improved heart function.

Interestingly, cardiac fibroblasts, isolated from osteopontin-null mice, maintained their ability to produce components of ECM but displayed altered proliferation and adhesion, suggesting that attenuated myocardial fibrosis in osteopontin-null mice in the Ang-II induced model of LV hypertrophy may be related to the cardiac fibroblast properties and not necessarily to the ECM synthesis [59]. Furthermore, a disarrayed collagen deposition has been demonstrated in the myocardium of osteopontin-null mice [71], suggesting a crucial role of osteopontin in ECM assembly and organization. Thus, in a number of HF models, the elevation of osteopontin levels in the myocardium is likely to exert beneficial effects by contributing to the formation of tissue-stabilizing fibrosis and supporting cardiomyocyte contractile function.

Deposition of insoluble collagen contributes to increased ECM accumulation and myocardial stiffness. The importance of posttranslational processing and deposition of collagen fibers in tissue fibrosis was demonstrated in TIMP-null mice subjected to Ang-II infusion, which displayed increased myocardial fibrosis along with significantly upregulated osteopontin expression, despite the lack of de novo synthesis of collagen type I [127]. Osteopontin has been shown to increase lysyl oxidase expression and activity, an enzyme that is responsible for the formation of cross-linked, insoluble collagen [84,90]. In the TAC model of LV hypertrophy, osteopontin expression was positively correlated with lysyl oxidase expression in the myocardium [84]. Pharmacological inhibition of osteopontin with ALK5 inhibitor SM16 attenuated myocardial fibrosis in TAC mice but was associated with LV dilatation, systolic dysfunction, and increased mortality [128]. These findings highlight the potential role of osteopontin in stabilizing ECM by modulating lysyl oxidase activity expression.

### 3.2.4. Osteopontin in Cardiac Capillarization

The angiogenic response is critical for scar formation and cardiac repair in different cardiac diseases [129]. In the remodeled myocardium, a proper level of nutrients and oxygen supply to enlarged cardiomyocytes is achieved by increased myocardial capillarization [12,13]. However, at some point in the course of the disease, the myocardium fails to maintain adequate tissue capillarization marking the transition of cardiac hypertrophy to HF [12,13]. The mechanisms controlling myocardial capillarization in cardiac hypertrophy and failure are not fully understood. Recent studies have identified various endogenous regulators of myocardial capillarization with complex interactions between various cell types in the heart [130]. Among these pro- and anti-angiogenic factors, osteopontin has also emerged as an essential regulator of myocardial angiogenesis.

Osteopontin contributes to angiogenesis by potentiating ILK and NF-κB-mediated hypoxia-inducible factor 1-α-dependent VEGF expression [131]. In the absence of osteopontin, myocardial angiogenesis is significantly impaired, resulting in adverse myocardial remodeling following MI [95]. The decrease in in vitro tube formation in cardiac endothelial cells isolated from osteopontin-null mice is restored by treatment with purified osteopontin [95]. Thus, osteopontin may play an important role in the cardiac remodeling following MI, at least in part, by preventing endothelial cell apoptosis, promoting endothelial cell regeneration, ultimately, and maintaining myocardial angiogenesis.

### 3.3. Clinical Implication of Osteopontin

Clinical studies have suggested that osteopontin might serve as a potent diagnostic and prognostic biomarker in diverse HF conditions. In this section, we discuss the roles and clinical implications of osteopontin in various HF diseases, including DCM, hypertensive HF, MI, and right HF (Figure 5).

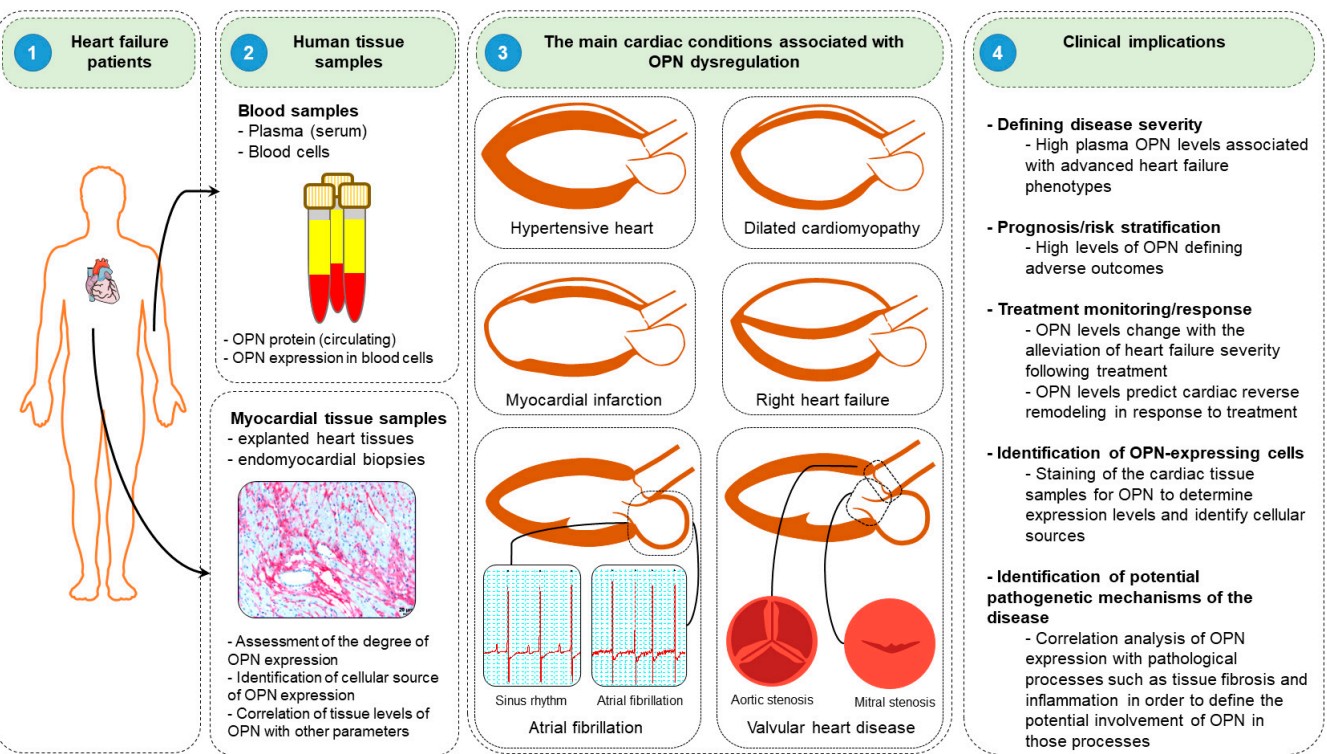

**Figure 5.** Clinical implications of osteopontin. (1) Blood samples are collected from the peripheral blood, while myocardial tissues are obtained from explanted hearts or endomyocardial biopsies during cardiac catheterization or cardiac surgery. (2) Plasma/serum and blood cells are isolated from blood samples to measure osteopontin (OPN) levels. Similarly, osteopontin expression levels are evaluated and cellular sources of osteopontin are identified in myocardial tissues. (3) Assessment of osteopontin levels might be useful in several cardiac conditions, including hypertensive heart, dilated cardiomyopathy, myocardial infarction, and right heart failure, atrial fibrillation and valvular heart disease. (4) Assessment of osteopontin levels might be useful in determining diseases severity, prognostic/risk stratification, monitoring of treatment response, identification of cellular sources of osteopontin in various cardiac conditions and help to understand the mechanisms of the disease pathogenesis.

### 3.3.1. Dilated Cardiomyopathy

DCM is a severe condition characterized by progressive ventricular chamber dilatation with underlying genetic mechanisms and is associated with increased myocardial inflammation and fibrosis [132]. Emerging evidence implicates osteopontin in the development and progression of DCM [81,113,133–139]. DCM patients display significantly increased myocardial osteopontin expression [81,113,133,134]. Moreover, myocardial osteopontin expression levels were correlated with the degree of myocardial fibrosis in DCM patients [134]. Analysis of the myocardial tissues from DCM patients revealed osteopontin expression in various cell types [113,133,135]. Immunohistochemistry and in situ hybridization of myocardial biopsies obtained from DCM patients demonstrated cardiomyocytes as the major source of increased osteopontin expression [135], although coronary vascular smooth muscles and cardiac fibroblasts were also found to express osteopontin [113,133,135]. There was a significant correlation between increased myocardial osteopontin expression levels and impaired hemodynamic parameters in DCM patients [113,133]. Moreover, plasma osteopontin levels reflected HF severity in DCM patients [136–138] and were associated with adverse outcomes [138,139]. Interestingly, LV assist device implantation in DCM patients was associated with attenuated myocardial osteopontin expression [135].

### 3.3.2. Hypertensive Heart Failure

Hypertensive heart disease is cardiac hypertrophy characterized by ventricular wall thickening in response to sustained blood pressure elevation. The main changes associated with cardiac hypertrophy include an increase in the size of individual cardiomyocytes, enhanced myocardial fibrosis, and alterations of the intramyocardial coronary vasculature. The mechanisms driving hypertrophic myocardial changes encompass not only the mechanical stress imposed on the ventricular wall by the elevated blood pressure but also the dysregulation of neurohormones, growth factors, inflammatory mediators, and matricellular proteins [24–26]. In several cardiac conditions associated with elevated intracardiac pressures, as seen in systemic hypertension or aortic stenosis in humans or rodent TAC models, myocardial osteopontin upregulation was associated with adverse changes in ECM homeostasis along with functional alterations of myocardial cells [29,58,59,67,68,84,127,140].

Patients with hypertensive HF display excessive myocardial accumulation of osteopontin, which is associated with increased LV stiffness and systolic dysfunction [90]. In hypertensive patients, circulating osteopontin levels are associated with both LV hypertrophy and LV diastolic dysfunction [141,142]. Increased circulating osteopontin levels are associated with adverse outcomes in patients with aortic stenosis [143]. Furthermore, elevated circulating osteopontin levels predict the irreversibility of LV hypertrophy after aortic valve replacement surgery [144].

### 3.3.3. Myocardial Infarction

MI is an acute coronary artery atherothrombotic occlusive disease characterized by myocardial cell death due to an imbalance between myocardial oxygen supply and demand [145]. In MI, ECM undergoes structural alterations following myocardial injury to facilitate the reparative processes in the heart [104]. Several studies implicated osteopontin in the ECM remodeling following MI [95,99,122]. In patients with MI, a significant relationship was revealed between plasma osteopontin levels and LV remodeling and dysfunction [146,147]. Circulating osteopontin levels could predict infarct size in patients with ST-elevated MI [147], and a combination of high osteopontin levels and high hs-CRP levels were significantly associated with increased risk of all-cause mortality, re-infarction, and HF [148].

Importantly, preclinical studies demonstrated adverse effects of complete abolition of osteopontin in this pathology. Thus, osteopontin-null mice exhibited more pronounced LV dilatation and systolic dysfunction following MI compared to wild-type counterparts [99,122]. In rodent hearts, osteopontin expression was increased in infarct as well as non-infarct regions post-MI [71]. Immunohistochemical analysis revealed positive staining for osteopontin, mainly in the interstitium in the infarct and non-infarct myocardium upon MI [71]. Subsequently, certain macrophage populations were identified as the main cell type responsible for the excessive osteopontin production in the myocardial interstitium following MI [63]. Osteopontin expression in macrophages was crucial for the manifestation of cardiac wound healing processes following experimental MI [100].

Accumulating evidence suggests that osteopontin plays an indispensable role in the myocardial healing processes after MI by regulating a variety of processes, including accumulation of apoptotic cells in the infarcted myocardium [99], attenuated myocardial fibrogenesis [122], augmented MMP-2 and MMP-9 activation [122], and impaired myocardial capillarization [95].

### 3.3.4. Atrial Fibrillation

Atrial fibrillation (AF) and HF frequently coexist and complicate one another [149,150], as both conditions share pathophysiological mechanisms and common risk factors [151]. Underlying mechanisms of AF involve structural remodeling characterized by atrial enlargement and tissue fibrosis [152,153]. In experimental murine models, osteopontin has recently been implicated in atrial fibrosis [154]. In line with these studies, significantly increased osteopontin plasma levels were revealed in AF patients with electrophysiologically

proven fibrosis compared to sinus rhythm controls [155,156]. In AF patients, atrial tissue expression of osteopontin was associated with augmented fibrosis [156].

Osteopontin has recently been identified as a novel independent risk marker for incident AF in a general Swedish population [157]. Furthermore, several studies demonstrated the role of osteopontin as a biomarker of treatment response and complications in AF patients. In patients undergoing cryoballoon ablation therapy for AF, elevated pre-procedure levels of osteopontin were related to AF recurrence [158]. In another study, plasma osteopontin levels were strongly associated with future ischemic stroke in AF patients during anticoagulant treatment [159]. Plasma osteopontin was also independently associated with major bleedings in AF patients on oral anticoagulants [160].

### 3.3.5. Valvular Heart Disease

Valvular heart disease represents one of the major causes of heart failure [161]. Osteopontin has been implicated in the regulation of mineral deposition in the cardiac valves, and alterations in osteopontin expression can lead to pathological valve changes [162]. Elevated circulating osteopontin levels were associated with the severity of valve calcification in patients with rheumatic mitral stenosis [163], CAD [164], and calcific aortic stenosis patients [143,165,166]. Notably, osteopontin levels were increased already in the pre-calcification stages of aortic valve degeneration [167]. Histological examinations of valve tissues obtained from patients with mitral stenosis [168] and patients undergoing aortic valve replacement surgery for aortic stenosis or regurgitation [169,170] demonstrated that osteopontin expression was associated with macrophage infiltration and calcium aggregation. In aortic stenosis patients, elevated circulating osteopontin levels were associated with a higher rate of atrial arrhythmia and increased risk of death during the follow-up compared to those with lower osteopontin levels [143]. Furthermore, high circulating osteopontin levels in these patients were associated with lower left ventricular hypertrophy regression after aortic valve replacement surgery [144].

### 3.3.6. Right Ventricular Failure

Several studies demonstrated elevated circulating osteopontin levels in PH patients with right HF [171,172]. In these patients, osteopontin levels correlated with a number of hemodynamic parameters such as pulmonary artery dispensability index [171], right atrial pressure [172], as well as functional parameters such as six-minute walking distance [171,172] and NYHA FC [172,173]. Importantly, baseline osteopontin levels were predictive of survival in PH patients [172–174]. In addition, circulating osteopontin levels were associated with maladaptive right HF in PH patients [171,175,176].

Preclinical studies demonstrated osteopontin upregulation in the RV in various rat models of PH, including monocrotaline injection [177,178], pulmonary artery banding [179], and exposure to SU5416/hypoxia [179] or to hypoxia alone [180]. Close dynamics of plasma osteopontin content and its expression levels in the remodeled RV myocardium suggested that RV was the main source of circulating osteopontin in monocrotaline [177] and hypoxia rats [180]. Interestingly, improvement in pulmonary hemodynamics and RV function by pharmacological agents such as peroxisome proliferator-activated receptor γ activator pioglitazone [181] and estrogen receptor-β agonist 17β-estradiol [177] was accompanied by a decrease in osteopontin expression. Although, in these studies, the decline in osteopontin expression might reflect diminished RV wall stress secondary to afterload reduction, direct effects of the agents on the RV cannot be excluded and need to be proven in studies employing afterload-independent models of right HF.

### 4. Osteopontin as a Potential Therapeutic Target in Heart Failure

Current evidence on osteopontin biology in cardiac homeostasis and diseases suggests that osteopontin might represent a potential therapeutic target in HF. It is now widely acknowledged that inhibition or stimulation of osteopontin function/expression and its upstream or downstream signaling pathways by different strategies such as neutraliz-

ing antibodies, small-molecular inhibitors, aptamers, and osteopontin peptide analogs might represent a promising strategy to modulate adverse cardiac remodeling in HF of various etiologies.

RNA aptamers are short (12–30 nt) RNA-based oligonucleotides capable of specifically binding to target proteins by forming a three-dimensional structure [182]. Aptamers are stable, lack immunogenicity, and are effective at very low concentrations, hence providing a substantial advantage over other inhibitory therapeutic alternatives [182]. Preventive administration of an osteopontin aptamer attenuated cardiac remodeling and dysfunction and reduced cardiomyocyte hypertrophy and cardiac fibrosis in the early (4 weeks) and late (12 weeks) stages of LV failure following TAC [126]. Similarly, inhibition of osteopontin by shRNA injected directly into the myocardium reduced cardiac remodeling and dysfunction in a mutant troponin mouse model of DCM [86]. Thus, therapeutic RNA molecules such as aptamers and shRNA can be a valuable tool in the therapeutic targeting of osteopontin in HF.

Several studies demonstrated the therapeutic effects of osteopontin neutralizing antibodies in certain models of HF. A polyclonal antibody M5Ab against a synthetic peptide (a cryptic epitope of osteopontin exposed by thrombin cleavage, VDVPNGRGDSLAYGLRS, M5 peptide) blocked profibrotic effects of osteopontin in a mouse liver fibrosis model [183]. Administration of neutralizing anti-osteopontin antibodies significantly mitigated HF severity in the genetic model of HF due to ILK deficiency [115]. Moreover, the application of osteopontin antibodies significantly reduced the myocardial expression of MMP-9, which is involved in the progression of HF in the MDX mouse model of DCM [123].

Some studies suggested that osteopontin fragments might have beneficial effects in some types of HF. Indeed, human osteopontin-derived peptide SVVYGLR induced angiogenesis in vitro and in vivo [184,185]. Moreover, the application of the human osteopontin isoform-C promoted neovascularization through activation and recruitment of macrophages in a murine ischemia-induced neo-vascularization model [36]. In a hamster model of cardiomyopathy, administration of the osteopontin-derived (SVVYGLR) SV peptide improved cardiac function and attenuated LV dilatation and cardiomyocyte hypertrophy [186].

Literature on the effects of currently available therapeutic approaches on osteopontin expression in HF patients is scarce. Reduction in plasma osteopontin levels in the early postoperative period (72 h) was reported in patients after coronary artery bypass grafting [187]. In another study, plasma levels and heart tissue expression of osteopontin were assessed in end-stage HF patients before and after LV assist device implantation and subsequent heart transplantation [135]. LV assist device support was not associated with any changes in plasma osteopontin levels in ischemic heart disease and DCM patients. In contrast, osteopontin mRNA expression in heart biopsy specimens decreased significantly after LV assist device support. Following heart transplantation, plasma osteopontin levels decreased significantly in all patients [135]. However, the importance of these findings needs further research.

## 5. Summary

This review has outlined the advances made toward our understanding of the biological functions of osteopontin in several cardiac pathologies. Identification of cellular sources of osteopontin has led to the discovery of many of its important cell-type-specific functions in diverse heart diseases. Osteopontin is expressed in the heart by diverse cell types, including cardiac endothelial cells, fibroblasts, macrophages, and cardiomyocytes. It regulates various processes such as inflammation, fibrogenesis, hypertrophy, and vascularization. The complex nature of the biological effects of osteopontin might account for the conflicting results obtained in different experimental models. The evidence summarized in this review suggests that the optimal expression of osteopontin is required to maintain tissue homeostasis in preventing or healing cardiac injuries, with a lack of osteopontin hampering the tissue injury and wound healing responses and osteopontin abundancy leading to adverse myocardial remodeling. In addition, a great number of studies shed light on the specific diagnostic and prognostic implications of circulating osteopontin in diverse cardiac conditions.

## 6. Current Challenges and Future Directions

The reports summarized in this review suggest that osteopontin plays a crucial role in the pathogenesis and clinical manifestations of cardiac pathologies. As most of the in vivo studies utilized global osteopontin-null mice, additional work is necessary to determine the molecular mechanisms of cell- and isoform-specific biologic functions of osteopontin. In vivo studies using cell-specific gain-off and loss-off function approaches in various HF models can help us gain a deeper understanding of cardiac cell-specific regulation of osteopontin signaling in specific disease settings. Future studies with "humanized" transgenic mouse models expressing specific human osteopontin isoforms are likely to advance this area of research, which eventually will provide enhanced translational insights into the physiologic and pathophysiologic roles of human osteopontin splice variants. All the available studies have focused only on the extracellular secreted form of osteopontin, and the intracellular role of osteopontin in heart diseases is still largely undetermined [188]. Similarly, the role of osteopontin in right HF remains unexplored as the majority of the studies investigated the role of osteopontin in left HF.

**Author Contributions:** Conceptualization, A.M., M.S. and A.S.; writing—original draft preparation, A.M., M.S. and A.S.; writing—review and editing, A.M., M.S., A.S.S., R.S. and A.S.; visualization, A.M. and A.S.; supervision, A.S.S., R.S. and A.S. All authors have read and agreed to the published version of the manuscript.

**Funding:** This research received no external funding.

**Institutional Review Board Statement:** Not applicable.

**Informed Consent Statement:** Not applicable.

**Data Availability Statement:** Not applicable.

**Conflicts of Interest:** The authors declare no conflict of interest.

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
