# Peer review of "Clinical and Molecular Implications of Osteopontin in Heart Failure"

_cimb, doi:10.3390/cimb44080245_

Round 1

Reviewer 1 Report

Dear Authors,

thank you for submitting your paper to Biology. I have read it with interest and I have the following comments:

- I have to congratulate with the depth of review provided and the inclusion of recent papers

- To put this review in clinical context, a missing piece of this review is how life-style, pharmacological and interventional/surgical procedures can revert or attenuate heart failure and how osteopontin behaves in the response to therapy. I sense that this paragraph will add tremendous value to you excellent work.

Author Response

We would like to thank the reviewer for careful evaluation and for the supportive words about our manuscript. We very much appreciate the reviewer´s valuable suggestions, which have been very helpful in improving the manuscript.

P1. To put this review in clinical context, a missing piece of this review is how life-style, pharmacological and interventional/surgical procedures can revert or attenuate heart failure and how osteopontin behaves in the response to therapy. I sense that this paragraph will add tremendous value to you excellent work.

R1. We thank the reviewer for the valuable comment. Unfortunately, literature on the effects of currently available therapeutic approaches on osteopontin expression in HF patients is scarce. Reduction in plasma osteopontin levels in the early post-operative period (72 h) was reported in patients after coronary artery by-pass grafting (Sbarouni et al. 2012). In another study, plasma levels and heart tissue expression of osteopontin were assessed in end-stage HF patients before and after LV assist device implantation and subsequent heart transplantation (Schipper et al. 2011). LV assist device support was not associated with any changes in plasma osteopontin levels in ischemic heart disease and DCM patients. In contrast, osteopontin mRNA expression in heart biopsy specimens decreased significantly after LV assist device support. Following heart transplantation, plasma osteopontin levels decreased significantly in all patients (Schipper et al. 2011). However, the importance of these findings needs further research.

We included this new information into the following subsection of the revised version of the manuscript “Osteopontin as a potential therapeutic target in heart failure”.

Reviewer 2 Report

The authors present an excellent and thorough review of osteopontin's clinical and molecular implications in heart failure.

I have several comments:

- The role of the atrium in heart failure development is currently being studied. It would be interesting to know whether osteopontin plays a role in this situation.

- Valve diseases are not discussed in the review. It would be interesting to mention whether there is any information on osteopontin in these clinical situations.  

- Finally, is there information on osteopontin and atrial fibrillation? 

Author Response

We would like to thank the reviewer for careful evaluation and for the supportive words about our manuscript. We very much appreciate the reviewer´s valuable suggestions, which have been very helpful in improving the manuscript.

P1. - The role of the atrium in heart failure development is currently being studied. It would be interesting to know whether osteopontin plays a role in this situation.

R1. We thank the reviewer for the suggestion. As pointed out by the reviewer, osteopontin plays a substantial role in atrial remodeling in HF. Atrial fibrillation was initially not within the scope of the review. Following reviewer’s recommendation, we have now summarized available literature on this topic and added a subsection with the title “atrial fibrillation” to the revised version of the manuscript. In addition, we have modified the figure 5 according to the new information in the revised version of the manuscript. We added the following text:

“Atrial fibrillation (AF) and HF frequently coexist and complicate one another (Santhanakrishnan et al. 2016, Bergau et al. 2022), as both conditions share pathophysiological mechanisms and common risk factors (Anter et al. 2009). Underlying mechanisms of AF involve structural remodeling characterized by atrial enlargement and tissue fibrosis (Lau et al. 2016, Cunha et al. 2022). In experimental murine models, osteopontin has recently been implicated in the atrial fibrosis (Du et al. 2022). In line with these studies, significantly increased osteopontin plasma levels were revealed in AF patients with electrophysiologically proven fibrosis compared to sinus rhythm controls (Li et al. 2020, Künzel et al. 2021). In AF patients, atrial tissue expression of osteopontin was associated with the augmented fibrosis (Lin et al. 2020).

Osteopontin has recently been identified as a novel independent risk marker for incident AF in a general Swedish population (Molvin et al. 2020). Furthermore, several studies demonstrated the role of osteopontin as a biomarker of treatment response and complications in AF patients. In patients undergoing cryoballoon ablation therapy for AF, elevated pre-procedure levels of osteopontin were related to AF recurrence (GüneÅŸ et al. 2017). In another study, plasma osteopontin levels were strongly associated with future ischemic stroke in AF patients during anticoagulant treatment (Hijazi et al. 2020). Plasma osteopontin was also independently associated with major bleedings in AF patients on oral anticoagulants (Siegbahn et al. 2021).”

P2. - Valve diseases are not discussed in the review. It would be interesting to mention whether there is any information on osteopontin in these clinical situations.

R2. We thank the reviewer for the suggestion. As noticed by the reviewer, osteopontin is also involved in the pathogenesis of valvular heart diseases. Accordingly, we have summarized available literature and discussed this topic and added a subsection with the title “valvular heart diseases” to the revised version of the manuscript. In addition, we have modified the figure 5 according to the new information made to the revised version of the manuscript. We added the following text:

“Valvular heart disease represents one of the major causes of heart failure (Adamo et al. 2020). Osteopontin has been implicated in the regulation of mineral deposition in the cardiac valves and alterations in osteopontin expression can lead to pathological valve changes (Lok and Lyle 2019). Elevated circulating osteopontin levels were associated with the severity of valve calcification in patients with rheumatic mitral stenosis (Atalar et al. 2006), CAD (Sponder et al. 2016) and calcific aortic stenosis patients (Yu et al. 2009, Ferrari et al. 2010, Lutz et al. 2017). Notably, osteopontin levels were increased already in pre-calcification stages of aortic valve degeneration (Grau et al. 2012). Histological examinations of valve tissues obtained from patients with mitral stenosis (Canver et al. 2000) and patients undergoing aortic valve replacement surgery for aortic stenosis or regurgitation (Kennedy et al. 2000, Breyne et al. 2010) demonstrated that osteopontin expression was associated with macrophage infiltration and calcium aggregation. In aortic stenosis patients, elevated circulating osteopontin levels were associated with higher rate of atrial arrhythmia and increased risk of death during the follow-up compared to those with lower osteopontin levels (Lutz et al. 2017). Furthermore, high circulating osteopontin levels in these patients were associated with lower left ventricular hypertrophy regression after aortic valve replacement surgery (Weber et al. 2020).”

P3. - Finally, is there information on osteopontin and atrial fibrillation?

R3. We thank the reviewer for the suggestion. We addressed this issue above (see R1

Round 2

Reviewer 1 Report

Dear Authors,

thank you for adding the paragraph that helped clarify also the future challenges for osteopontin research